# Model-Agnostic Meta-Attack: Towards Reliable Evaluation of Adversarial Robustness

## Abstract

The vulnerability of deep neural networks to adversarial examples has motivated an increasing number of defense strategies for promoting model robustness. However, the progress is usually hampered by insufficient robustness evaluations. As the *de facto* standard to evaluate adversarial robustness, adversarial attacks typically solve an optimization problem of crafting adversarial examples with an iterative process. In this work, we propose a Model-Agnostic Meta-Attack (MAMA) approach to discover stronger attack algorithms *automatically*. Our method learns the optimizer in adversarial attacks parameterized by a recurrent neural network, which is trained over a class of data samples and defenses to produce effective update directions during adversarial example generation. Furthermore, we develop a model-agnostic training algorithm to improve the generalization ability of the learned optimizer when attacking unseen defenses. Our approach can be flexibly incorporated with various attacks and consistently improves the performance with little extra computational cost. Extensive experiments demonstrate the effectiveness of the learned attacks by MAMA compared to the state-of-the-art attacks on different defenses, leading to a more reliable evaluation of adversarial robustness.

## 1 Introduction

Deep neural networks are vulnerable to maliciously crafted adversarial examples (Biggio et al., 2013; Szegedy et al., 2014; Goodfellow et al., 2015), which aim to induce erroneous model predictions by adding small perturbations to the normal inputs. Due to the threats, a multitude of defense strategies have been proposed to improve adversarial robustness (Kurakin et al., 2017b; Madry et al., 2018; Liao et al., 2018; Wong & Kolter, 2018; Zhang et al., 2019b; Dong et al., 2020a; Pang et al., 2020). However, many defenses have later been shown to be ineffective due to incomplete or incorrect robustness evaluations (Athalye et al., 2018a; Uesato et al., 2018; Carlini et al., 2019; Croce & Hein, 2020b; Dong et al., 2020b; Tramer et al., 2020), making it particularly challenging to understand their effects and identify the actual progress of the field. Therefore, developing methods that can evaluate adversarial robustness accurately and reliably is of profound importance.

Adversarial attacks are broadly adopted as an indispensable solution to evaluate adversarial robustness for different models. One of the most prominent attacks is the *projected gradient descent* (PGD) method (Madry et al., 2018), which generates an adversarial example by performing iterative gradient updates to maximize a classification loss (e.g., cross-entropy loss) w.r.t. the input. Although recent methods improve upon PGD by introducing different loss functions (Carlini & Wagner, 2017; Gowal et al., 2019; Croce & Hein, 2020b; Sriramanan et al., 2020) and adjusting the step size (Croce & Hein, 2020b), most of them adopt hand-designed optimization algorithms, such as vanilla gradient descent, momentum (Polyak, 1964), and Adam (Kingma & Ba, 2015). However, it has been shown that these attacks can be sub-optimal (Croce & Hein, 2020b; Tramer et al., 2020), arousing an overestimation of adversarial robustness for some defenses. It is thus imperative to develop more effective optimization algorithms for improving adversarial attacks. Nevertheless, designing a generic optimization algorithm in a hand-crafted manner is non-trivial, considering the different defense models with varying network architectures, defense strategies, and training datasets.

To overcome this issue and develop stronger adversarial attacks, in this paper, we propose a **Model-Agnostic Meta-Attack (MAMA)** approach to *learn optimization algorithms* in adversarial attacks *automatically*. In particular, we parameterize the optimizer with a recurrent neural network (RNN)

to mimic the behavior of iterative attacks, which outputs an update direction at each time step during adversarial example generation. By learning to solve a class of optimization problems with different data samples and defense models, the RNN optimizer could benefit from long-term dependencies of iterative attacks and exploit common structures among the optimization problems. To improve and stabilize training, we then propose a *prior-guided refinement strategy*, which imposes a prior given by the PGD update rule on the outputs of the RNN optimizer, whose parameters are trained via the maximum a posteriori (MAP) estimation framework. Consequently, the RNN optimizer can learn to yield more effective update directions than hand-designed ones.

Despite the effectiveness, the learned optimizer may not generalize well to attacking *unseen* defenses due to potential overfitting to the defenses used for training. Although training a different optimizer for each defense is possible, the training process can be time-consuming and inconvenient in practice when it is utilized to benchmark adversarial robustness. To endow the learned optimizer with a better generalization ability, we develop a *model-agnostic training algorithm* with a gradient-based meta-train and meta-test process to simulate the shift from seen to unseen defenses. Therefore, the learned optimizer can directly be deployed to attack unseen defenses without retraining or fine-tuning.

Extensive experiments validate the effectiveness and generalization ability of the learned optimizers for attacking different defense models. We also demonstrate the flexibility of our method integrating with various attacks, including those with different loss functions (Carlini & Wagner, 2017; Gowal et al., 2019; Croce & Hein, 2020b) and initialization (Tashiro et al., 2020). Our method consistently improves the attacking performance over baselines, while introducing little extra computational cost. By incorporating our method with an orthogonal technique — output diversified initialization (ODI) (Tashiro et al., 2020), we achieve lower robust test accuracy on all 12 defense models that we study than other state-of-the-art attacks, including the MultiTargeted (MT) attack (Gowal et al., 2019) and AutoAttack (Croce & Hein, 2020b), leading to a more reliable evaluation of adversarial robustness. Moreover, MAMA provides a plug-and-play module in adversarial attacks, which may benefit future adversarial attacks if new improvements have been made in different aspects.

## 2 PRELIMINARIES AND RELATED WORK

### 2.1 ADVERSARIAL ATTACKS

Let $f(\boldsymbol{x}) : \boldsymbol{x} \in \mathbb{R}^D \rightarrow \boldsymbol{z} \in \mathbb{R}^K$ denote a classifier, which outputs the logits $\boldsymbol{z} := [z_1, ..., z_K]$ over $K$ classes for an input image $\boldsymbol{x}$. The prediction of $f$ is $C_f(\boldsymbol{x}) = \arg\max_{i=1,...,K} z_i$. Given the true label $y$ of $\boldsymbol{x}$, adversarial attacks aim to generate an adversarial example $\hat{\boldsymbol{x}}$ that is misclassified by $f$ (i.e., $C_f(\hat{\boldsymbol{x}}) \neq y$), while the distance between the adversarial example $\hat{\boldsymbol{x}}$ and the natural one $\boldsymbol{x}$ measured by the $\ell_\infty$ norm is smaller than a threshold $\epsilon$ as $\|\hat{\boldsymbol{x}} - \boldsymbol{x}\|_\infty \leq \epsilon$. Although we introduce our approach based on the $\ell_\infty$ norm only, the extension to other $\ell_p$ norms is straightforward. The adversarial example $\hat{\boldsymbol{x}}$ can be generated by solving a constrained optimization problem as

$$\hat{\boldsymbol{x}} = \arg\max_{\boldsymbol{x}'} \mathcal{L}(f(\boldsymbol{x}'), y), \quad \text{s.t. } \|\boldsymbol{x}' - \boldsymbol{x}\|_\infty \leq \epsilon, \tag{1}$$

where $\mathcal{L}$ is a loss function on top of the classifier $f(\boldsymbol{x})$. For example, $\mathcal{L}$ could be the cross-entropy loss as $\mathcal{L}_{ce}(f(\boldsymbol{x}), y) = -\log p_y$ where $p_y = e^{z_y}/\sum_{i=1}^K e^{z_i}$ denotes the predicted probability of class $y$, or the margin-based CW loss (Carlini & Wagner, 2017) as $\mathcal{L}_{cw}(f(\boldsymbol{x}), y) = \max_{i \neq y} z_i - z_y$.

A lot of gradient-based methods (Goodfellow et al., 2015; Kurakin et al., 2017a; Carlini & Wagner, 2017; Madry et al., 2018; Dong et al., 2018) have been proposed to solve this optimization problem. The projected gradient descent (PGD) method (Madry et al., 2018) performs iterative updates as

$$\hat{\boldsymbol{x}}_{t+1} = \Pi_{\mathcal{B}_\epsilon(\boldsymbol{x})}\big(\hat{\boldsymbol{x}}_t + \alpha_t \cdot \text{sign}(\nabla_{\boldsymbol{x}} \mathcal{L}(f(\hat{\boldsymbol{x}}_t), y))\big), \tag{2}$$

where $\mathcal{B}_\epsilon(\boldsymbol{x}) = \{\boldsymbol{x}' : \|\boldsymbol{x}' - \boldsymbol{x}\|_\infty \leq \epsilon\}$ denotes the $\ell_\infty$ ball centered at $\boldsymbol{x}$ with radius $\epsilon$, $\Pi(\cdot)$ is the projection operation, and $\alpha_t$ is the step size at the $t$-th attack iteration. PGD initializes $\hat{\boldsymbol{x}}_0$ randomly by uniform sampling within $\mathcal{B}_\epsilon(\boldsymbol{x})$ and adopts a fixed step size $\alpha$ in every iteration.

To develop stronger attacks for reliable robustness evaluation, recent improvements upon PGD include adopting different loss functions (Gowal et al., 2019; Croce & Hein, 2020b; Sriramanan et al., 2020), adjusting the step size (Croce & Hein, 2020b), and designing sampling strategies of initialization (Tashiro et al., 2020). For the update rules, besides the vanilla gradient descent adopted in PGD, recent attacks (Dong et al., 2018; Gowal et al., 2019) suggest to use the momentum (Polyak, 1964)

and Adam (Kingma & Ba, 2015) optimizers. In contrast, we aim to enhance adversarial attacks by learning optimization algorithms in an automatic way.

## 2.2 ADVERSARIAL DEFENSES

To build robust models against adversarial attacks, numerous defense strategies have been proposed, but most defenses can be evaded by new adaptive attacks (Athalye et al., 2018a; Tramer et al., 2020). Among the existing defenses, adversarial training (AT) is arguably the most effective defense technique, in which the network is trained on the adversarial examples generated by attacks. Based on the primary AT frameworks like PGD-AT (Madry et al., 2018) and TRADES (Zhang et al., 2019b), improvements have been made via ensemble learning (Tramèr et al., 2018; Pang et al., 2019), metric learning (Mao et al., 2019; Pang et al., 2020), semi-supervised learning (Alayrac et al., 2019; Carmon et al., 2019; Zhai et al., 2019), and self-supervised learning (Chen et al., 2020a; Hendrycks et al., 2019; Kim et al., 2020). Due to the high computational cost of AT, other efforts are devoted to accelerating the training procedure (Shafahi et al., 2019; Wong et al., 2020; Zhang et al., 2019a). Recent works emphasize the training tricks (e.g., weight decay, batch size, etc.) in AT (Gowal et al., 2020; Pang et al., 2021). However, it has been shown that the robust test accuracy of numerous adversarially trained models can be degraded significantly by using stronger attacks (Croce & Hein, 2020b; Tramer et al., 2020), indicating that the reliable evaluation of adversarial robustness remains an imperative yet challenging task. In this paper, we focus on evaluating adversarial robustness of AT models due to their superior robustness over other defense techniques.

## 2.3 META-LEARNING

Meta-learning (learning to learn) studies how to learn learning algorithms automatically from a meta perspective, and has shown promise in few-shot learning (Finn et al., 2017) and learning optimizers (Andrychowicz et al., 2016). The latter is called learning to optimize (L2O). The first L2O method leverages a coordinate-wise long short term memory (LSTM) network as the optimizer model, which is trained by gradient descent on a class of optimization problems (Andrychowicz et al., 2016). The optimizer can also be trained by reinforcement learning (Li & Malik, 2017). Bello et al. (2017) propose to adopt an RNN controller to generate the update rules of optimization algorithms, which can be described as a domain specific language. L2O is then extended to efficiently optimize black-box functions (Chen et al., 2017). Recent methods focus on improving the training stability and generalization of the learned optimizers by designing better optimizer models and training techniques (Lv et al., 2017; Wichrowska et al., 2017; Chen et al., 2020b).

The general idea of meta-learning has been investigated in adversarial attacks. Most related methods learn to generate adversarial examples by using convolutional neural networks (CNNs), which usually take clean images as inputs and return adversarial examples/perturbations (Baluja & Fischer, 2017; Poursaeed et al., 2018). Differently, our method adopts an RNN model as the attack optimizer to mimic the behavior of iterative attacks. Empirical results validate the superiority of our approach compared to the CNN-based generators. Meta-learning has also been used in black-box attacks (Du et al., 2020) and AT defenses (Xiong & Hsieh, 2020; Jiang et al., 2021). It is noteworthy that Xiong & Hsieh (2020) propose a similar approach to ours that uses an RNN optimizer to learn attacks for AT. Although the RNN optimizer is also adopted in our approach, we improve the attacking performance and generalization ability of the RNN optimizer with new techniques, as presented in Sec. 3. More detailed comparisons between our work and Xiong & Hsieh (2020) are shown in Appendix A.

## 3 METHODOLOGY

In this paper, we propose a **Model-Agnostic Meta-Attack (MAMA)** approach to improve adversarial attacks by learning optimization algorithms automatically. MAMA permits learning to exploit common structures of the optimization problems (1) over a class of input samples and defense models. We first present the instantiation of the learned optimizer with a recurrent neural network (RNN) and the corresponding training procedure in Sec. 3.1. The learned optimizer is expected to not only perform well on the data samples and defenses for which it is trained but also possess an excellent **generalization ability** to other unseen data samples, defenses, and more optimization steps (i.e., attack iterations) than training. Therefore, we propose a model-agnostic training algorithm simulating train/test shift to ensure generalization of the learned optimizer, as detailed in Sec. 3.2. An overview of our proposed MAMA approach is provided in Fig. 1.

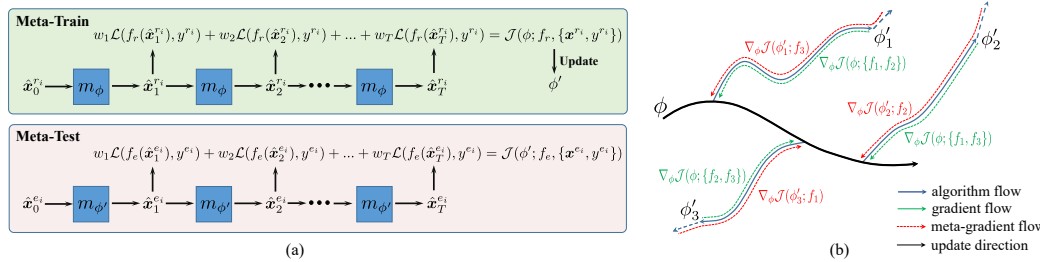

Figure 1: An overview of the MAMA approach. (a) shows the architecture of meta-attack with an RNN optimizer. (b) represents the model-agnostic optimization process. As an example, given defense models $\{f_1, f_2, f_3\}$, meta-train/meta-test could include different divisions as $\{f_1, f_2\}/\{f_3\}$, $\{f_1, f_3\}/\{f_2\}$, or $\{f_2, f_3\}/\{f_1\}$. The parameters $\phi$ of the optimizer will be updated by accumulating the gradients from meta-train and meta-gradients from meta-test.

### 3.1 META-ATTACK WITH RECURRENT NEURAL NETWORKS

Different from the existing attack algorithms, we propose a *meta-attack* that learns an optimizer in adversarial attacks parameterized by an RNN model $m_\phi$ with parameters $\phi$ to mimic the behavior of iterative attacks. The learned optimizer is capable of optimizing the loss function $\mathcal{L}$ in Eq. (1) for different data samples $(\boldsymbol{x}, y)$ and defense models $f$. At each step, $m_\phi$ takes gradient information of the loss function (1) at the current adversarial example as input and outputs an update direction for generating the adversarial example. This procedure can be expressed as

$$\hat{\boldsymbol{x}}_{t+1} = \Pi_{\mathcal{B}_\epsilon(\boldsymbol{x})}\left(\hat{\boldsymbol{x}}_t + \alpha_t \cdot \boldsymbol{g}_t\right), \quad [\boldsymbol{g}_t, \boldsymbol{h}_{t+1}] = m_\phi\left(\nabla_t, \boldsymbol{h}_t\right), \quad (3)$$

where $\alpha_t$ is the step size, $\nabla_t = \nabla_{\boldsymbol{x}}\mathcal{L}(f(\hat{\boldsymbol{x}}_t), y)$ denotes the gradient of the loss function $\mathcal{L}$ at $\hat{\boldsymbol{x}}_t$, and $\boldsymbol{h}_t$ denotes the hidden state representation of the optimizer, which contains rich temporal gradient information to capture long-term dependencies. The output $\boldsymbol{g}_t$ of the RNN model is used to update the adversarial example $\hat{\boldsymbol{x}}_t$. We apply a $\tanh$ transformation to the last layer of the RNN model to normalize and smoothen the update direction $\boldsymbol{g}_t$ since the adversarial perturbations are required to be smaller than the threshold $\epsilon$. In particular, we adopt the coordinate-wise long short term memory (LSTM) network (Andrychowicz et al., 2016) as the RNN model $m_\phi$, which can significantly reduce trainable parameters, making training much easier and faster.

Given a distribution $\mathcal{F}$ over defenses $f$ and a data distribution $\mathcal{D}$ over $(\boldsymbol{x}, y)$ pairs, the parameters $\phi$ can be trained by maximizing the expected loss over the entire trajectory of optimization as

$$\mathcal{J}(\phi; \mathcal{F}, \mathcal{D}) = \mathbb{E}_{f \sim \mathcal{F}, (\boldsymbol{x}, y) \sim \mathcal{D}}\left[\sum_{t=1}^{T} w_t \mathcal{L}(f(\hat{\boldsymbol{x}}_t), y)\right], \quad (4)$$

where $\hat{\boldsymbol{x}}_t$ is obtained by the optimizer model $m_\phi$ through Eq. (3), $w_t \in \mathbb{R}_{\geq 0}$ is the weight for the loss at the $t$-th step to emphasize different importance on the loss at different steps, and $T$ is the total number of optimization steps. For example, we could let $w_t = \mathbf{1}[t = T]$ to focus on the loss at the final adversarial example since it would be used to attack the classifier, but using a different setting for $w_t$ can make training easier. For training the optimizer network $m_\phi$, we can maximize the objective (4) using gradient ascent on $\phi$. The expectation in Eq. (4) is approximated by mini-batches of randomly sampled defense models $f$ and data points $(\boldsymbol{x}, y)$. The gradient w.r.t. $\phi$ is calculated by truncated backpropagation. Besides the general framework of learning to optimize (Andrychowicz et al., 2016; Chen et al., 2020b; Xiong & Hsieh, 2020), there exist particular challenges in training the optimizer due to the optimization problem of adversarial attacks as illustrated below.

**Vanishing gradient problem.** Note that the optimization problem (1) is a constrained one, which is different from unconstrained problems commonly studied in previous works (Andrychowicz et al., 2016; Chen et al., 2020b; Lv et al., 2017; Wichrowska et al., 2017). As in Eq. (3), we need to project the adversarial example onto the $\ell_\infty$ ball $\mathcal{B}_\epsilon(\boldsymbol{x})$ after every optimization step, which is essentially the *clip* function. However, optimizing Eq. (4) requires backpropagation through the clip function, which can result in a number of zero gradient values when the update exceeds the $\ell_\infty$ ball. This can consequently lead to the vanishing gradient problem for a long optimization trajectory. To address this problem, we apply the straight-through estimator (Bengio et al., 2013) to get an approximate gradient of the clip function as $\nabla_{\boldsymbol{x}'}\Pi_{\mathcal{B}_\epsilon(\boldsymbol{x})}(\boldsymbol{x}') = I$, where $I$ denotes the identity matrix. In this way, the optimizer can effectively be trained to optimize the constrained problem (1).

---

**Algorithm 1** Model-Agnostic Meta-Attack (MAMA) algorithm

---

**Require:** A data distribution $\mathcal{D}$ over $(\boldsymbol{x}, y)$ pairs, a set of defense models $\mathcal{F} = \{f_j\}_{j=1}^N$, the number of defenses $n$ in the meta-test set, the objective function $\mathcal{J}$ in Eq. (5), attack iterations $T$, batch size $B$, learning rates $\beta, \gamma$, balancing hyperparameter $\mu$.
**Ensure:** The parameters $\phi$ of the learned optimizer.
1: **for** iter = 1 **to** MaxIterations **do**
2:      Randomly draw $N - n$ defenses from $\mathcal{F}$ to get $\hat{\mathcal{F}}$, and let $\bar{\mathcal{F}} = \mathcal{F}\backslash\hat{\mathcal{F}}$ consist of the other $n$ defenses;
3:      **for** $f_r$ in $\hat{\mathcal{F}}$ **do**                                                             ▷ Meta-train
4:          Randomly draw a mini-batch $\mathcal{B}_r = \{(\boldsymbol{x}^{r_i}, y^{r_i})\}_{i=1}^B$ from $\mathcal{D}$;
5:          For $(\boldsymbol{x}^{r_i}, y^{r_i}) \in \mathcal{B}_r$, compute $\hat{\boldsymbol{x}}_t^{r_i}$ from $\phi$ for defense $f_r$ by Eq. (3) for $t = 1$ to $T$;
6:      **end for**
7:      Compute $\mathcal{J}(\phi; \hat{\mathcal{F}}, \mathcal{D})$ using the defenses $f_r$ in $\hat{\mathcal{F}}$ and the corresponding mini-batches $\mathcal{B}_r$;
8:      $\phi' = \phi + \beta\nabla_\phi\mathcal{J}(\phi; \hat{\mathcal{F}}, \mathcal{D})$;
9:      **for** $f_e$ in $\bar{\mathcal{F}}$ **do**                                                             ▷ Meta-test
10:          Randomly draw a mini-batch $\mathcal{B}_e = \{(\boldsymbol{x}^{e_i}, y^{e_i})\}_{i=1}^B$ from $\mathcal{D}$;
11:          For $(\boldsymbol{x}^{e_i}, y^{e_i}) \in \mathcal{B}_e$, compute $\hat{\boldsymbol{x}}_t^{e_i}$ from $\phi'$ for defense $f_e$ by Eq. (3) for $t = 1$ to $T$;
12:      **end for**
13:      Compute $\mathcal{J}(\phi'; \bar{\mathcal{F}}, \mathcal{D})$ using the defenses $f_e$ in $\bar{\mathcal{F}}$ and the corresponding mini-batches $\mathcal{B}_e$;
14:      $\phi \leftarrow \phi + \gamma\nabla_\phi(\mathcal{J}(\phi; \hat{\mathcal{F}}, \mathcal{D}) + \mu\mathcal{J}(\phi'; \bar{\mathcal{F}}, \mathcal{D}))$;
15: **end for**

---

**Training instability problem.** Training the learned optimizer may suffer from an instability problem (Chen et al., 2020b), leading to unsatisfactory performance. To further stabilize training of the learned optimizer $m_\phi$ and improve its performance, we propose a *prior-guided refinement strategy* which leverages an informative prior to refine the outputs of the optimizer. In particular, the prior is given by the PGD update rule (Madry et al., 2018) which performs generally well on many defenses. We specify the prior distribution of $\boldsymbol{g}_t$ to be an isotropic Gaussian distribution as $\mathcal{N}(\text{sign}(\nabla_t), \frac{1}{2\lambda_t})$, whose mean is the PGD update direction $\text{sign}(\nabla_t)$. The variance of the distribution is $1/2\lambda_t$ with $\lambda_t$ being a hyperparameter. Under the maximum a posteriori (MAP) estimation framework, the objective function (4) can be viewed as the likelihood while the prior is equivalent to minimizing the squared $\ell_2$ distance as $\lambda_t\|\boldsymbol{g}_t - \text{sign}(\nabla_t)\|_2^2$. We provide a formal derivation of MAP in Appendix B. Therefore, the overall objective function of training the optimizer becomes

$$\mathcal{J}(\phi; \mathcal{F}, \mathcal{D}) = \mathbb{E}_{f\sim\mathcal{F}, (\boldsymbol{x}, y)\sim\mathcal{D}}\left[\sum_{t=1}^T \left(w_t\mathcal{L}(f(\hat{\boldsymbol{x}}_t), y) - \lambda_t\|\boldsymbol{g}_t - \text{sign}(\nabla_t)\|_2^2\right)\right], \quad (5)$$

where $\{\lambda_t\}_{t=1}^T$ can be seen as another set of weights corresponding to the regularization terms at each step. In this way, the optimizer can learn from the informative prior given by the PGD update rule and explore potential better directions simultaneously. To maximize the objective function (5), a basic technique is using gradient ascent on $\phi$. We call it **Basic Meta-Attack (BMA)**.

## 3.2 Model-agnostic meta-attack algorithm

To make robustness evaluation more convenient after training, the learned optimizer should generalize well to new data samples, defense models, and more optimization steps that are *not seen* during training. In practice, we observe that the data-driven process of optimizing Eq. (5) (i.e., BMA) enables to obtain a generalizable optimizer on new data samples with more optimization steps, but the learned optimizer may not necessarily generalize well to new defenses, as shown in Sec. 4.3. This is because the defenses usually adopt different network architectures and defense strategies, making it hard for an attack optimizer to generalize well across a variety of defenses (Tramer et al., 2020).

To improve the generalization ability of the learned optimizer when attacking unseen defenses, we propose a model-agnostic training algorithm for meta-attack (i.e., **MAMA**). This method simulates the shift between defenses during training with a gradient-based meta-train and meta-test procedure, inspired by model-agnostic meta-learning (Finn et al., 2017; Li et al., 2018). After MAMA training, the learned optimizer can directly be used to attack unseen defenses, which will be more convenient and user-friendly for carrying out robustness evaluation once novel defense methods are presented.

Specifically, assume that we have a set of $N$ defenses (denoted as $\mathcal{F}$) in training, where we reuse the notation $\mathcal{F}$ without ambiguity. To simulate the shift between defense models in training and testing,

Table 1: Classification accuracy (%) of four defense models on **CIFAR-10** against various *white-box* attacks with different optimizers, including vanilla gradient descent (PGD), momentum (MI-FGSM), nesterov accelerated gradient (NI-FGSM), Adam, Adv-CNN, and the learned optimizer (BMA).

| Defense | PGD | MI-FGSM | NI-FGSM | Adam | Adv-CNN | BMA (Ours) |
|---|---|---|---|---|---|---|
| HYDRA | 58.73 | 59.27 | 59.61 | 58.69 | 59.42 | **58.57** |
| Pre-training | 56.77 | 57.12 | 57.20 | 56.77 | 57.40 | **56.63** |
| Overfitting | 54.27 | 54.67 | 54.85 | 54.30 | 54.95 | **54.13** |
| FastAT | 46.69 | 47.33 | 47.30 | 46.69 | 47.58 | **46.54** |

at each training iteration we split $\mathcal{F}$ into a *meta-train* set $\hat{\mathcal{F}}$ of $N - n$ defenses and a *meta-test* set $\bar{\mathcal{F}}$ of the other $n$ defenses. The model-agnostic training objective is to maximize the loss $\mathcal{J}(\phi; \hat{\mathcal{F}}, \mathcal{D})$ on the meta-train defenses, while ensuring that the update direction of $\phi$ also leads to maximizing $\mathcal{J}(\phi; \bar{\mathcal{F}}, \mathcal{D})$ on the meta-test defenses, such that the learned optimizer will perform well on true test defenses. During meta-train, the optimizer parameters $\phi$ are updated by the one-step gradient ascent

$$\phi' = \phi + \beta \nabla_\phi \mathcal{J}(\phi; \hat{\mathcal{F}}, \mathcal{D}),$$

where $\beta$ is the meta-train learning rate. During meta-test, we expect the updated parameters $\phi'$ to exhibit a good generalization performance on the meta-test defenses, which indicates that $\mathcal{J}(\phi'; \bar{\mathcal{F}}, \mathcal{D})$ is also maximized. The meta-train and meta-test can be optimized simultaneously as

$$\max_\phi \mathcal{J}(\phi; \hat{\mathcal{F}}, \mathcal{D}) + \mu \mathcal{J}(\phi + \beta \nabla_\phi \mathcal{J}(\phi; \hat{\mathcal{F}}, \mathcal{D}); \bar{\mathcal{F}}, \mathcal{D}), \tag{6}$$

where $\mu$ is a hyperparameter to balance the meta-train loss and the meta-test loss. The objective (6) is optimized by gradient ascent. After accumulating the gradients from meta-train and meta-gradients from meta-test, the optimizer is trained to perform well on attacking both meta-train and meta-test defenses. The gradient and meta-gradient flows in the computation graph are illustrated in Fig. 1(b). We summarize the overall MAMA algorithm in Alg. 1.

## 4 EXPERIMENTS

In this section, we present the experimental results to validate the effectiveness of BMA and MAMA for adversarial attacks. We follow a simple-to-complex procedure to conduct experiments. Sec. 4.2 first shows the advantages of the learned optimizer in BMA when considering one defense at a time. Sec. 4.3 then proves that the learned optimizer possesses good cross-data, cross-iteration, and cross-model generalization abilities, respectively. Sec. 4.4 demonstrates that BMA and MAMA can achieve state-of-the-art performance by comparing with various top-performing attacks on 12 defense models. More details about the baseline attacks and defenses are provided in Appendix C and D, respectively. We mainly present the results based on the $\ell_\infty$ norm in this section. The results based on the $\ell_2$ norm and standard deviations of multiple runs are provided in Appendix E.

### 4.1 EXPERIMENTAL SETUP

The hyperparameters of baseline attacks are set according to the common practice or their original implementations. Specifically, in Sec. 4.2 and Sec. 4.3, we set the attack iterations as $T = 20$ and the fixed step size as $\alpha = 2/255$ according to (Madry et al., 2018). The learned optimizer in our method also adopts the same setting for fair comparison. In Sec. 4.4, we adopt the official hyperparameter configurations of the baseline attacks since we assume that their hyperparameters have been tuned appropriately. Our BMA and MAMA attacks follow a similar setup with 100 iterations, 20 random restarts, and ODI initialization. In all experiments, we use a two-layer LSTM with 20 hidden units in each layer as the optimizer model. We set $w_t = 1$ and $\lambda_t = 0.1$ in Eq. (5) and study different $\lambda_t$ through an ablation study. We set the learning rates as $\beta = 0.0001$ and $\gamma = 0.001$ in MAMA with $\mu = 1.0$. The mini-batch size is $B = 32$.

### 4.2 EFFECTIVENESS AND ADAPTIVENESS OF BMA

We first train a different attack optimizer for each defense by BMA. We consider four typical AT defenses on CIFAR-10 (Krizhevsky & Hinton, 2009) — HYDRA (Sehwag et al., 2020), Pre-Training (Hendrycks et al., 2019), Overfitting (Rice et al., 2020), and FastAT (Wong et al., 2020).

**Comparison with hand-designed optimizers.** Since most existing adversarial attacks adopt hand-designed optimizers, we first compare the performance of our learned optimizer with these hand-designed ones. Specifically, we adopt the CW loss (Carlini & Wagner, 2017) as $\mathcal{L}$ in Eq. (1) and set

Table 2: Classification accuracy (%) of four defenses on **CIFAR-10** against PGD$_{CE}$, PGD$_{DLR}$, MT, ODI-R10, APGD$_{DLR}$ and their extensions by integrating with our proposed BMA method. The default attack iterations are set as $T = 20$ as detailed in Sec. 4.1, and we also involve another 100-step APGD$_{DLR}$ as APGD$_{DLR}$-100.

| Defense | Fixed Step Size | | | | | | | | Automatic Step Size | | | |
|---|---|---|---|---|---|---|---|---|---|---|---|---|
| | PGD$_{CE}$ | | PGD$_{DLR}$ | | MT | | ODI-R10 | | APGD$_{DLR}$ | | APGD$_{DLR}$-100 | |
| | Baseline | +BMA | Baseline | +BMA | Baseline | +BMA | Baseline | +BMA | Baseline | +BMA | Baseline | +BMA |
| HYDRA | 60.38 | **60.27** | 59.16 | **59.02** | 57.63 | **57.54** | 57.50±0.01 | **57.41±0.01** | 58.89 | **58.74** | 58.47 | **58.37** |
| Pre-training | 57.89 | **57.69** | 57.86 | **57.66** | 55.16 | **55.09** | 55.11±0.01 | **55.04±0.01** | 57.83 | **57.44** | 57.44 | **57.15** |
| Overfitting | 56.08 | **55.87** | 55.29 | **55.13** | 55.71 | **55.64** | 52.68±0.02 | **52.57±0.01** | 55.19 | **54.84** | 54.60 | **54.45** |
| FastAT | 46.82 | **46.70** | 48.19 | **48.01** | 43.85 | **43.72** | 43.76±0.01 | **43.66±0.01** | 48.02 | **47.53** | 47.18 | **46.85** |

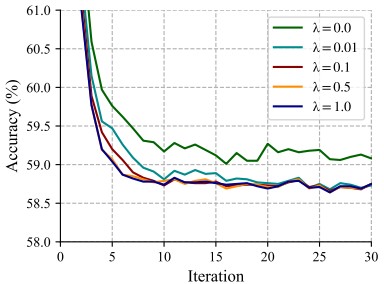

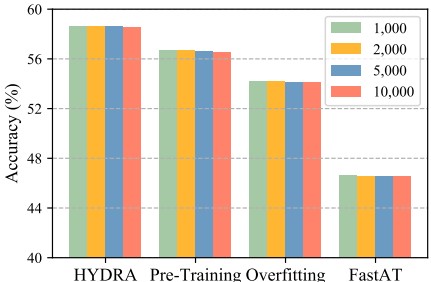

Figure 2: The accuracy curves of HYDRA w.r.t. training iterations of BMA with different $\lambda$.

Figure 3: Cross-data generalization evaluation with different numbers of training data points on **CIFAR-10**.

$\epsilon = 8/255$. We compare our learned optimizer (in BMA) with vanilla gradient descent (used in PGD (Madry et al., 2018)), momentum (Polyak, 1964) (used in MI-FGSM (Dong et al., 2018)), nesterov accelerated gradient (Nesterov, 1983) (used in NI-FGSM (Lin et al., 2019)), and Adam (Kingma & Ba, 2015) (used in (Uesato et al., 2018; Gowal et al., 2019)). Besides, we also adopt a CNN-based adversarial example generation method (Adv-CNN) (Baluja & Fischer, 2017) for comparison. We report the classification accuracy of the four defenses against these attacks in Table 1. We can see that BMA with the learned optimizer leads to lower accuracy of the defense models than the existing attacks, demonstrating that BMA can discover stronger adversarial attacks automatically.

**Ablation study of the prior-guided refinement strategy.** We conduct an ablation study to investigate the effects of the prior-guided refinement strategy introduced in Sec. 3.1. Fig. 2 shows the accuracy of HYDRA along the training iterations of BMA with $\lambda_t := \lambda = 0.0, 0.01, 0.1, 0.5$, and 1.0, respectively. After integrating the PGD update rule as a prior with $\lambda > 0$, the learned optimizer achieves faster convergence and better performance than that with $\lambda = 0$. The results indicate that the proposed prior-guided refinement strategy can benefit from the informative prior given by the PGD update rule and explore potential better update directions simultaneously. Moreover, as $\lambda$ increases, to a certain extent the convergence tends to be faster, but the performance tends to be similar. Therefore, we select $\lambda = 0.1$ for all experiments in the training phase of BMA.

**Adaptiveness.** The learned optimizer is generally compatible with previous methods on improving adversarial attacks in different aspects. To verify this, we incorporate our method with PGD variants using the cross-entropy loss (PGD$_{CE}$) (Madry et al., 2018), the difference of logits ratio loss (PGD$_{DLR}$) (Croce & Hein, 2020b) and PGD$_{DLR}$ with varying step size (APGD$_{DLR}$) (Croce & Hein, 2020b), the MultiTarged (MT) attack (Gowal et al., 2019) that performs targeted attack for each class, and the output diversified initialization method (Tashiro et al., 2020) with 10 random restarts (ODI-R10). Note that these methods only change the loss function $\mathcal{L}$ or the starting point $\hat{x}_0$, which do not affect the learning process of BMA. Table 2 shows the results of the baseline methods and their extensions by integrating with the learned optimizer in BMA. The results verify that integrating BMA can consistently enhance the performance of each method. The mechanism of combining BMA is concise and easy to implement, yet steadily improving the different attack methods.

### 4.3 GENERALIZATION EVALUATION

**Cross-data generalization.** We first evaluate the cross-data generalization ability of the learned optimizer in BMA on unseen data points. Specifically, we randomly select $1,000$, $2,000$, $5,000$, and all $10,000$ data points from the CIFAR-10 test set as the training set for BMA, respectively, and adopt the entire test set of CIFAR-10 for evaluation. The performance on the entire test set is shown in Fig. 3. We can see that the learned optimizers using different numbers of data points achieve

Table 3: Cross-model generalization performance of four unseen defenses on **CIFAR-10**, while the optimizer is trained on four random defenses from TRADES (T), AWP (A), Unlabeled (U), Hydra (H), and SAT (S).

| Training | Pre-Training | | | BoT | | | PGD-AT | | | FastAT | | |
|---|---|---|---|---|---|---|---|---|---|---|---|---|
| Defenses | BMA$_{bs}$ | BMA$_{en}$ | MAMA | BMA$_{bs}$ | BMA$_{en}$ | MAMA | BMA$_{bs}$ | BMA$_{en}$ | MAMA | BMA$_{bs}$ | BMA$_{en}$ | MAMA |
| {T,A,U,H} | 56.84 | 56.79 | **56.74** | 56.01 | 55.98 | **55.93** | 50.82 | 50.80 | **50.75** | 46.77 | 46.75 | **46.69** |
| {T,A,U,S} | 56.84 | 56.81 | **56.74** | 56.02 | 55.98 | **55.94** | 50.82 | 50.80 | **50.74** | 46.78 | 46.75 | **46.69** |
| {T,U,H,S} | 56.85 | 56.80 | **56.75** | 56.02 | 55.99 | **55.93** | 50.82 | 50.81 | **50.76** | 46.78 | 46.75 | **46.68** |

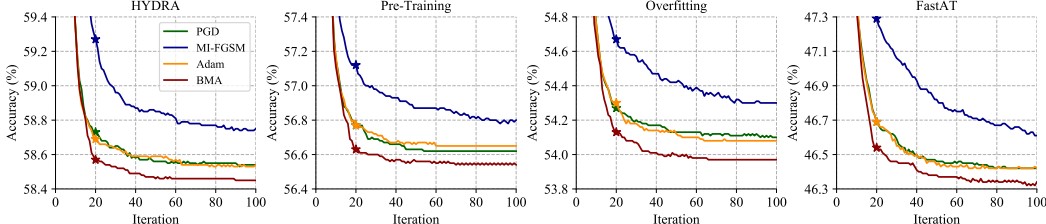

Figure 4: Comparing the performance of different optimizers based on PGD adversary against four defense models. The iteration needed for training the automated optimizer (BMA) is also marked at 20. The results also evidence that BMA can directly extend attack iterations from 20 to a longer step.

similar attack performance on the four defense models, indicating that the data-driven procedure enables the learned optimizer to obtain a good generalization ability to new unseen data samples.

**Cross-iteration generalization.** We then evaluate the generalization ability of the learned optimizer in BMA to more attack iterations than training. Fig. 4 shows the curves of model accuracy w.r.t. the number of attack iterations ranging from 1 to 100. We also mark the number of iterations used for training BMA at 20. The results on four defense models verify that BMA can consistently outperform the baseline attacks with varying attack iterations. It demonstrates that the learned optimizers also have a good generalization ability across attack iterations.

**Cross-model generalization.** The training process may face inconvenience in practice when training a different optimizer for each defense. A better property of the learned optimizer is more generalizable for adversarial attacks on any unseen defenses. We thus consider the challenging and practical scenario — training the optimizer on several available defense models and evaluating its generalization ability to other unseen defense models. This setting requires the optimizer to learn more general knowledge for adversarial attacks. For choosing the defense models in training, we mainly consider using a set of diverse defenses. Thus we randomly sample four models to train the optimizer from TRADES (T) (Zhang et al., 2019b) , AWP (A) (Wu et al., 2020), Unlabeled (U) (Carmon et al., 2019), Hydra (H) (Sehwag et al., 2020), and SAT (S) (Huang et al., 2020). MAMA randomly selects three defenses as meta-train and the other one as meta-test in each iteration. We compare MAMA with two BMA variants. The first one trains an optimizer for each training defense and chooses the best optimizer for evaluation (BMA$_{bs}$), and the second one optimizes Eq. (5) using the training defenses as an ensemble (BMA$_{en}$). Table 3 shows that MAMA is more effective than the two methods, indicating that the generalization ability of the optimizer can be improved by adopting the model-agnostic algorithm. The effect of meta-test components is listed in Appendix E.3.

## 4.4 COMPARISON WITH STATE-OF-THE-ART ATTACKS

In this section, we perform a large-scale experiment to compare the performance of our methods with various state-of-the-art attacks for robustness evaluation of 12 defense models on CIFAR-10. We consider both **BMA** that learns a different attack optimizer for each defense model and **MAMA** that only learns one generalizable optimizer adopting the model-agnostic training algorithm to test all defenses. We compare our methods with several typical and state-of-the-art attacks, including 1) **PGD$_{CE}$** (Madry et al., 2018), 2) **PGD$_{CW}$**, 4) **APGD** (Croce & Hein, 2020b) with the DLR loss, 3) **MT**, 5) AutoAttack (**AA**) (Croce & Hein, 2020b), and 6) **ODI** (Tashiro et al., 2020). PGD$_{CE}$, PGD$_{CW}$, and ODI use 100 iterations with step size $2/255$ and 20 restarts. APGD also uses the same iterations and restarts. MT performs targeted attacks for 9 classes except the true one with 2 restarts and 100 attack iterations. We adopt the official implementation of AA. We integrate our methods with ODI to establish stronger attacks. Our methods also use 100 attack iterations with 20 ODI restarts. MAMA is trained on defense models {T,A,U,H} with the model-agnostic algorithm.

Table 4 shows the classification accuracy of these defense models against different adversarial attacks. We also show the clean accuracy and the **reported** robust accuracy of these models in the

Table 4: Classification accuracy (%) of different defenses on **CIFAR-10** under various *white-box* attacks. The defenses marked with [†] adopt additional unlabeled datasets. The defenses marked with [*] adopt $\epsilon = 0.031$ as originally reported, and the others adopt $\epsilon = 8/255$. $\mathbf{\Delta}$ represents the difference between reported accuracy and testing accuracy by ours. $\mathbf{\Delta}'$ represents the difference between accuracy under the best attack and ours. We also present the complexity (forward times) of each attack method in *blue*.

| Defense | Clean | Reported | PGD$_{CE}$ | PGD$_{CW}$ | APGD | MT | AA | ODI | BMA | MAMA | $\mathbf{\Delta}$ | $\mathbf{\Delta}'$ |
|---|---|---|---|---|---|---|---|---|---|---|---|---|
| | | | *2k* | *2k* | *2k* | *1.8k* | *7.4k* | *2k* | *2k* | *2k* | | |
| AWP[†] | 88.25 | 60.04 | 63.12 | 60.50 | 60.46 | 60.17 | 60.04 | 60.10 | 59.99 | **59.98** | -0.06 | -0.06 |
| Unlabeled[†] | 89.69 | 62.50 | 61.69 | 60.61 | 60.55 | 59.73 | 59.53 | 59.65 | **59.50** | **59.50** | -3.00 | -0.03 |
| HYDRA[†] | 88.98 | 59.98 | 59.53 | 58.19 | 58.20 | 57.34 | 57.20 | 57.24 | **57.11** | 57.12 | -2.87 | -0.09 |
| Pre-Training[†] | 87.11 | 57.40 | 57.07 | 56.36 | 56.90 | 55.00 | 54.94 | 54.96 | 54.82 | **54.81** | -2.59 | -0.13 |
| BoT | 85.14 | 54.39 | 56.15 | 55.53 | 55.66 | 54.44 | 54.39 | 54.34 | **54.24** | **54.24** | -0.15 | -0.15 |
| AT-HE | 85.14 | 62.14 | 61.72 | 55.27 | 55.87 | 53.81 | 53.74 | 53.79 | **53.69** | 53.70 | -8.45 | -0.05 |
| Overfitting | 85.34 | 58.00 | 56.87 | 55.10 | 55.60 | 53.43 | 53.42 | 53.42 | 53.38 | **53.37** | -4.63 | -0.05 |
| SAT[*] | 83.48 | 58.03 | 56.21 | 54.39 | 54.54 | 53.46 | 53.34 | 53.42 | 53.28 | **53.27** | -4.76 | -0.07 |
| TRADES[*] | 84.92 | 56.43 | 55.22 | 53.95 | 53.94 | 53.25 | 53.09 | 53.09 | **53.01** | 53.02 | -3.42 | -0.07 |
| LS | 86.09 | 53.00 | 55.81 | 54.44 | 54.90 | 52.82 | 53.00 | 52.72 | 52.61 | **52.60** | -0.40 | -0.12 |
| YOPO | 87.20 | 47.98 | 45.98 | 46.88 | 47.19 | 44.97 | 44.87 | 44.94 | **44.81** | 44.82 | -3.17 | -0.06 |
| FastAT | 83.80 | 46.44 | 45.93 | 45.88 | 46.37 | 43.53 | 43.41 | 43.50 | **43.29** | 43.32 | -3.15 | -0.12 |

Table 5: Classification accuracy (%) of different defenses on **CIFAR-100** and **ImageNet** under various *white-box* attacks. We directly apply the learned optimizer in MAMA on CIFAR-10 without further training.

| Defense | Dataset | $\epsilon$ | Clean | Reported | PGD$_{CE}$ | AA | MAMA | $\mathbf{\Delta}$ | $\mathbf{\Delta}'$ |
|---|---|---|---|---|---|---|---|---|---|
| AWP | | 8/255 | 60.38 | 28.86 | 33.27 | 28.86 | **28.85** | -0.01 | -0.01 |
| Pre-Training[†] | *CIFAR-100* | 8/255 | 59.23 | 33.50 | 33.10 | 28.42 | **28.38** | -5.12 | -0.04 |
| Overfitting | | 8/255 | 53.83 | 28.10 | 20.50 | 18.96 | **18.92** | -9.18 | -0.04 |
| FreeAT-HE (ResNet-50) | *ImageNet* | 4/255 | 61.83 | 39.85 | 36.70 | 30.65 | **30.60** | -9.25 | -0.05 |
| FreeAT-HE (WRN-50-2) | | 4/255 | 65.28 | 43.47 | 42.61 | **32.65** | 32.65 | -10.82 | 0.00 |

corresponding papers. Our methods achieve lower robust test accuracy on all 12 defense models than the baseline attacks, including MT and AA, demonstrating that ours can discover stronger adversarial attacks automatically than the existing attacks with hand-designed optimizers. Meanwhile, our methods are not only stronger but also less complex than AA. Besides, the results also demonstrate that MAMA can achieve a comparable performance of BMA, showing that the learned optimizer has a good generalization ability to unseen defense models due to the involved model-agnostic training algorithm, and could be flexibly used to evaluate the robustness of future defenses.

**Robustness evaluation on other datasets.** We *directly* apply the learned optimizer in MAMA on CIFAR-10 to evaluate adversarial robustness on CIFAR-100 (Krizhevsky & Hinton, 2009) and ImageNet (Deng et al., 2009), without updating the optimizer by new datasets or defenses. We adopt $\epsilon = 8/255$ on CIFAR-100 and $\epsilon = 4/255$ on ImageNet. During testing, there is a significant shift between training and testing defense models for MAMA. Despite these difficulties, MAMA can consistently improve the performance on most defenses in Table 5. The results verify that MAMA obtains a good generalization ability for adversarial attacks on unseen datasets, benefiting from the data-driven learning procedure automatically.

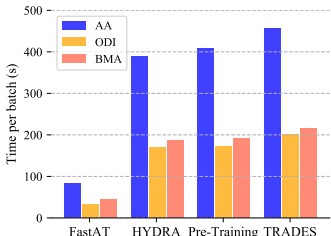

Figure 5: The running time of AA, ODI, and BMA with a mini-batch of 100 data points on **CIFAR-10**.

**Running time**. Fig. 5 shows the running time of AA, ODI, and BMA (MAMA) based on a single RTX 2080-Ti GPU. BMA is slightly slower than the baseline ODI, but significantly faster than AA.

## 5 CONCLUSION

In this paper, we propose a Model-Agnostic Meta-Attack (MAMA) approach to automatically discover stronger attack algorithms. Specifically, we learn an optimizer in adversarial attacks parameterized by an RNN model, which is trained over a class of data samples and defense models to produce effective update directions for crafting adversarial examples. We further develop a model-agnostic learning approach with a gradient-based meta-train and meta-test procedure to improve the generalization ability of the learned optimizer to unseen defense models. Extensive experimental results demonstrate the effectiveness, adaptiveness, and generalization ability of MAMA on different defense models, leading to a more reliable evaluation of adversarial robustness.

ETHICS STATEMENT

Our work aims at evaluating the adversarial robustness of defense models more accurately and reliably. It could identify the security problems of deep learning models before they are deployed to avoid potential harm caused by adversarial examples. We are unaware of any direct negative impact of this work. One limitation of our work is that the proposed methods (i.e., BMA and MAMA) cannot directly be used to attack defenses with obfuscated gradients. We still need to design some adaptive attack strategies (e.g., Expectation over Transformation (Athalye et al., 2018b) for randomized defenses) to apply our methods.

REPRODUCIBILITY STATEMENT

We provide the code for reproducing the results in the supplementary material.

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

## A    MORE DISCUSSIONS ON RELATED WORK

Xiong & Hsieh (2020) propose a similar approach to ours that uses an RNN optimizer to learn attacks for adversarial training. The differences between our method and Xiong & Hsieh (2020) are three-fold. First, the motivations are different: we propose to discover stronger adversarial attack algorithms while they aim to learn robust models by adversarial training. Second, based on the different goals, the techniques are different: we develop a prior-guided refinement strategy and a model-agnostic training algorithm to improve the attacking performance and generalization ability of the RNN optimizer while they do not adopt these techniques and use a relatively smaller number of attack iterations (i.e., $T = 10$). Third, the empirical evaluations are different: we mainly conduct experiments to show the attack performance on various defense models while they mainly consider the defense performance of adversarial training. Overall, our method differs from Xiong & Hsieh (2020) in the aspects of motivation, technique, and experiment. Despite using a similar Learning-to-Learn framework, our method has its own contributions in developing an effective attack method for evaluating adversarial robustness.

## B    EXPLANATION OF MAXIMUM A POSTERIORI

Assume that the output of the RNN optimizer is a random variable $g_t$. The objective of the optimizer is to maximize the loss $\mathcal{L}$ at $\hat{x}_t$ after applying $g_t$, as in Eq. (3). By using the cross-entropy loss as $\mathcal{L}$, we actually want to minimize $p_y$ (i.e., the probability of the true class) given $\hat{x}_t$. We let $p(y|g_t) = \frac{1}{p_y}$ such that we want to maximize $p(y|g_t)$. Thus $p(y|g_t)$ can be viewed as the conditional likelihood. By assuming the prior distribution of $g_t$ to be a Gaussian one as $p(g_t) \sim \mathcal{N}(\text{sign}(\nabla_t), \frac{1}{2\lambda_t})$, we can estimate the posterior distribution of $g_t$ using Bayes' theorem

$$p(g_t|y) \sim p(y|g_t)p(g_t).$$

Under the maximum a posteriori (MAP) framework, we would like to solve

$$\arg\max \log p(g_t|y) = \arg\max\{-\log p_y - \lambda_t\|g_t - \text{sign}(\nabla_t)\|_2^2\}$$
$$= \arg\max\{\mathcal{L}(f(\hat{x}_t), y) - \lambda_t\|g_t - \text{sign}(\nabla_t)\|_2^2\}.$$

Be accumulating the loss at each step, we can obtain Eq. (5).

## C    TECHNICAL DETAILS OF ADVERSARIAL ATTACKS

In this section, we introduce more related backgrounds and technical details for reference.

**PGD.** One of the most commonly studied adversarial attacks is the projected gradient descent (PGD) method (Madry et al., 2018). PGD iteratively crafts the adversarial example as

$$\hat{x}_{t+1} = \Pi_{\mathcal{B}_\epsilon(x)}\big(\hat{x}_t + \alpha_t \cdot \text{sign}(\nabla_x\mathcal{L}(f(\hat{x}_t), y))\big),$$

where $\mathcal{B}_\epsilon(x) = \{x' : \|x' - x\|_\infty \le \epsilon\}$ denotes the $\ell_\infty$ ball centered at $x$ with radius $\epsilon$, $\Pi(\cdot)$ is the projection operation, and $\alpha_t$ is the step size at the $t$-th attack iteration. $\mathcal{L}$ could be the cross-entropy loss as $\mathcal{L}_{ce}(f(x), y) = -\log p_y$ where $p_y = e^{z_y}/\sum_{i=1}^{K} e^{z_i}$ denotes the predicted probability of class $y$, or the margin-based CW loss (Carlini & Wagner, 2017) as $\mathcal{L}_{cw}(f(x), y) = \max_{i \ne y} z_i - z_y$.

**MI-FGSM.** Dong et al. (2018) propose to adopt the momentum optimizer (Polyak, 1964) in adversarial attacks, which can be formulated as

$$g_{t+1} = \mu \cdot g_t + \frac{\nabla_x\mathcal{L}(f(\hat{x}_t), y)}{\|\nabla_x\mathcal{L}(f(\hat{x}_t), y)\|_1}; \ \hat{x}_{t+1} = \Pi_{\mathcal{B}_\epsilon(x)}(\hat{x}_t + \alpha \cdot \text{sign}(g_{t+1})),$$

where $\alpha$ is a fixed step size and $\mu$ is a hyperparameter.

**NI-FGSM.** Lin et al. (2019) propose to adopt the Nesterov accelerated gradient (Nesterov, 1983) in adversarial attacks, as

$$x_t^{nes} = \hat{x}_t + \alpha \cdot \mu \cdot g_t; \ g_{t+1} = \mu \cdot g_t + \frac{\nabla_x\mathcal{L}(f(x_t^{nes}), y)}{\|\nabla_x\mathcal{L}(f(x_t^{nes}), y)\|_1}; \hat{x}_{t+1} = \Pi_{\mathcal{B}_\epsilon(x)}(\hat{x}_t + \alpha \cdot \text{sign}(g_{t+1})).$$

**Adam.** Some attacks (Uesato et al., 2018; Gowal et al., 2019) propose to adopt the Adam optimizer (Kingma & Ba, 2015) in adversarial attacks. By replacing the sign gradient in PGD by the Adam optimization rule, we can get the attack based on Adam easily.

**Adv-CNN.** Baluja & Fischer (2017) first propose to generate adversarial examples against a target classifier by training feed-forward CNN networks. To efficiently craft adversarial examples under white-box attack, we directly feed the gradients of adversarial images into the CNN, and output the update direction as

$$\boldsymbol{g}'_t = \mathrm{G}\left(\mathcal{L}(f(\hat{\boldsymbol{x}}_t), y)\right); \quad \hat{\boldsymbol{x}}_{t+1} = \Pi_{\mathcal{B}_\epsilon(\boldsymbol{x})}\left(\hat{\boldsymbol{x}}_t + \alpha_t \cdot \boldsymbol{g}'_t\right),$$

where G indicates the feed-forward CNN.

**AutoAttack (AA).** Croce & Hein (2020b) first propose the Auto-PGD (APGD) algorithm, which can automatically tune the adversarial step sizes along with the optimization trend. As for the adversarial objective, they develop a new difference of logits ratio (DLR) loss as

$$\mathcal{L}_{DLR}(f(\boldsymbol{x}), y) = -\frac{z_y - \max_{i \neq y} z_i}{z_{\pi_1} - z_{\pi_3}},$$

where $z$ is the logits and $\pi$ is the ordering which sorts the components of $z$. Finally, the authors propose to group $\text{APGD}_{\text{CE}}$ and $\text{APGD}_{\text{DLR}}$ with FAB (Croce & Hein, 2020a) and square attack (Andriushchenko et al., 2020) to form the AutoAttack (AA).

**ODI attack.** The Output Diversified Sampling (ODS) (Tashiro et al., 2020) belongs to a sampling strategy that aims to maximize diversity in the models' outputs when crafting adversarial examples. The formulation of ODI attack derived from ODS can be described as

$$\hat{\boldsymbol{x}}_{k+1} = \Pi_{\mathcal{B}_\epsilon(\boldsymbol{x})}\left(\hat{\boldsymbol{x}}_k + \alpha_t \cdot \text{sign}(\text{v}_{\text{ODS}}(\hat{\boldsymbol{x}}_k, f, \text{w}_d))\right), \ \text{v}_{\text{ODS}}(\hat{\boldsymbol{x}}_k, f, \text{w}_d) = \frac{\nabla_{\boldsymbol{x}}(\text{w}_d^T f(\hat{\boldsymbol{x}}_k))}{\|\nabla_{\boldsymbol{x}}(\text{w}_d^T f(\hat{\boldsymbol{x}}_k))\|_2},$$

where $\text{v}_{\text{ODS}}(\cdot)$ generates output-diversified starting points, and $\text{w}_d$ is sampled from the uniform distribution over $[-1, 1]^C$.

**MultiTarged (MT) attack.** MT (Gowal et al., 2019) performs targeted attack for all other classes except the true one, which iteratively crafts the adversarial example as

$$\hat{\boldsymbol{x}}_{t+1} = \Pi_{\mathcal{B}_\epsilon(\boldsymbol{x})}\left(\hat{\boldsymbol{x}}_t - \alpha_t \cdot \text{sign}(\nabla_{\boldsymbol{x}}\mathcal{L}(f(\hat{\boldsymbol{x}}_t), y^c))\right),$$

where $y^c$ belongs to the specific targeted attack class.

## D REFERENCE CODE OF ADVERSARIAL DEFENSES

We provide the architectures and code links for the defenses used in our paper in Table 6. Please note that the checkpoints of evaluating adversarial robustness are released in their code implementations.

Table 6: We summarize the architectures and code links for the referred defense methods.

| Method | Architecture | Code link |
|---|---|---|
| AWP (Wu et al., 2020) | WRN-28-10 | github.com/csdongxian/AWP |
| Unlabeled (Carmon et al., 2019) | WRN-28-10 | github.com/yaircarmon/semisup-adv |
| HYDRA (Sehwag et al., 2020) | WRN-28-10 | github.com/inspire-group/hydra |
| Pre-Training (Hendrycks et al., 2019) | WRN-28-10 | github.com/hendrycks/ss-ood |
| BoT (Pang et al., 2021) | WRN-34-20 | github.com/P2333/Bag-of-Tricks-for-AT |
| AT-HE (Pang et al., 2020) | WRN-34-20 | github.com/ShawnXYang/AT_HE |
| Overfitting (Rice et al., 2020) | WRN-34-20 | github.com/locuslab/robust_overfitting |
| SAT* (Huang et al., 2020) | WRN-34-10 | github.com/LayneH/self-adaptive-training |
| TRADES* (Zhang et al., 2019b) | WRN-34-10 | github.com/yaodongyu/TRADES |
| LS (Pang et al., 2021) | WRN-34-10 | github.com/P2333/Bag-of-Tricks-for-AT |
| YOPO (Zhang et al., 2019a) | WRN-34-10 | github.com/a1600012888/YOPO-You-Only-Propagate-Once |
| FastAT (Wong et al., 2020) | ResNet18 | github.com/locuslab/fast_adversarial |
| FreeAT-HE (Pang et al., 2020) | ResNet-50 | github.com/ShawnXYang/AT_HE |

## E MORE EXPERIMENTS

### E.1 EFFECTIVENESS UNDER THE $\ell_2$ NORM

Apart from the results based on the $\ell_\infty$ norm, we study the effectiveness based on the $\ell_2$ norm. The performance of our learned optimizer with these hand-designed ones is presented in this sec-

tion. Specifically, we adopt the CW loss (Carlini & Wagner, 2017) and set $\epsilon = 0.1$. We compare our learned optimizer (in BMA) with vanilla gradient descent (used in PGD (Madry et al., 2018)), momentum (Polyak, 1964) (used in MI-FGSM (Dong et al., 2018)), nesterov accelerated gradient (Nesterov, 1983) (used in NI-FGSM (Lin et al., 2019)), and Adam (Kingma & Ba, 2015) (used in (Gowal et al., 2019; Uesato et al., 2018)). For fair comparison, we set the attack iterations as $T = 20$ and the fixed step size as $\alpha = 0.025$ in all methods. The learned optimizer in BMA is trained with 20 attack iterations on the entire test set of CIFAR-10. We report the classification accuracy of the two defenses including Overfitting (Rice et al., 2020) and Robustness (Engstrom et al., 2019) against these attacks in Table 7. BMA with the learned optimizer leads to lower accuracy of the defense models than the existing attacks with hand-designed optimizers, demonstrating the effectiveness of BMA based on the $\ell_2$ norm.

Table 7: Classification accuracy (%) of defense models on **CIFAR-10** against various *white-box* attacks with different optimizers, including PGD, MI-FGSM, NI-FGSM, Adam and the learned optimizer (BMA), based on the $\ell_2$ norm.

| Defense | PGD | MI-FGSM | NI-FGSM | Adam | BMA (Ours) |
|---|---|---|---|---|---|
| Overfitting | 68.80 | 69.07 | 69.02 | 76.56 | **68.64** |
| Robustness | 70.29 | 70.46 | 70.39 | 78.52 | **70.04** |

### E.2 RANDOMNESS EXPERIMENTS

To avoid the randomness caused by a single experiment, we report the standard deviations of multiple runs on CIFAR-10 in Table 8. We can see that the learned optimizers using different runs achieve similar attack performance on the four defense models including HYDRA (Sehwag et al., 2020), Pre-training (Hendrycks et al., 2019), Overfitting (Rice et al., 2020) and FastAT (Wong et al., 2020). The results also demonstrate the stability of BMA to existing attacks.

Table 8: Standard deviations of multiple runs on **CIFAR-10** against $PGD_{CW}$ and its extension be integrating with our proposed BMA method.

| Method | Baseline | +BMA | +BMA (run-1) | +BMA (run-2) | +BMA (run-3) | +BMA (run-4) |
|---|---|---|---|---|---|---|
| HYDRA | 58.73 | **58.58**$\pm$ 0.01 | 58.57 | 58.59 | 58.58 | 58.57 |
| Pre-training | 56.77 | **56.64**$\pm$ 0.01 | 56.63 | 56.63 | 56.65 | 56.64 |
| Overfitting | 54.27 | **54.14**$\pm$ 0.01 | 54.15 | 54.13 | 54.14 | 54.13 |
| FastAT | 46.69 | **46.54**$\pm$ 0.01 | 46.54 | 46.55 | 46.55 | 46.54 |

### E.3 ESTIMATE THE EFFECT OF META-TESTING COMPONENT

**Effect of different $\mu$.** The meta-train and meta-test can be optimized simultaneously as Eq. (6), and $\mu$ is the hyperparameter to balance the meta-train and meta-test losses. To estimate the effect of different $\mu$, we sample four models to train the optimizer from TRADES (T), AWP (A), Unlabeled (U), Hydra (H), and SAT (S), and test the performance on Pre-Training for different $\mu$. We can see in Table 9, a proper value 1.0 gives the best performance, indicating that the meta-train and meta-test should be equally learned.

Table 9: Testing the performance for different $\mu$.

| $\mu$ | 0.5 | 0.8 | 1 | 1.2 | 1.5 |
|---|---|---|---|---|---|
| Accuracy (%) | 56.79 | 56.77 | 56.74 | 56.76 | 56.77 |

**Effect of different numbers of meta-train and meta-test.** Apart from the results based on different hyparameters, we study the effectiveness for different numbers of meta-train and meta-test. In the experiments, we attempt to sample $m$ domains as meta-train, and remaining $n$ domains as meta-test, thus named Sm-Tn. We report the classification accuracy of the defense model on different numbers of domains in Table 10. We can see that the best performance can be obtained by setting $m$ as 3 and $n$ as 1, which is also adopted in all experiments of MAMA.

Table 10: Testing the performance of different numbers of meta-train and meta-test.

| Sampling mode | S1T3 | S2T2 | S3T1 |
|---|---|---|---|
| Accuracy (%) | 56.80 | 56.78 | 56.74 |

### E.4 AUTOMATIC STEP SIZE

We also attempt to apply the automatic step size scheduler (Croce & Hein, 2020b) to MAMA in Table 11, which adopts 100 iterations and 20 restarts based on ODI on the different defenses including TRADES (Zhang et al., 2019b), Pre-training (Hendrycks et al., 2019), HYDRA (Sehwag et al., 2020). The results show that integrating the automatic step size does not further improve the overall performance for robustness evaluation. We think that combining MAMA with ODI can already converge to sufficiently good results for robustness evaluation and may not be improved by integrating other techniques such as automatic step size.

Table 11: Classification accuracy (%) of defense models on CIFAR-10 against MAMA with fixed or automatic step sizes.

| Method | Fixed Step Size | Automatic Step Size |
|---|---|---|
| TRADES | 53.02 | 53.01 |
| Pre-Training | 54.81 | 54.81 |
| HYDRA | 57.12 | 57.11 |

### E.5 OTHER TYPES OF DEFENSES

We further conduct experiments by considering other types of defenses rather than adversarial training. Specifically, we include a certified defense — CROWN-IBP (Zhang et al., 2020) and a randomized defense — JEM-1 (Grathwohl et al., 2020). We compare our proposed methods BMA and MAMA with AutoAttack (AA), which is the most effective baseline. We re-train the attack optimizer for each model in BMA and directly use the learned optimizer (i.e., the same as Table 4) in MAMA. To counter the randomness of JEM-1, we average the gradients over 3 runs (known as Expectation over Transformation (EoT) (Athalye et al., 2018b)). The results are shown in Table 12. Our methods consistently achieve better performance than AutoAttack.

Table 12: Classification accuracy (%) of different types of defense models on CIFAR-10 against various white-box attacks.

| Defense | AA | BMA | MAMA |
|---|---|---|---|
| CROWN-IBP | 33.71 | **33.60** | 33.64 |
| JEM-1 | 7.40 | **6.35** | 6.57 |

