# OpenReview forum: "Model-Agnostic Meta-Attack: Towards Reliable Evaluation of Adversarial Robustness"
_ICLR.cc/2022/Conference — ICLR 2022 Submitted_

### Official Review · Reviewer_jTw1 · 2021-10-22

**Correctness:** 4
**Technical Novelty And Significance:** 2
**Empirical Novelty And Significance:** 2
**Recommendation:** 6
**Confidence:** 2

**Main Review:**

The presentation of the paper was very good. Clearly laid out and comprehensively presented, with thorough experiments to back up the claims.

I am not an expert in this particular area, and so my comments may be misguided or have missed key issues (my Confidence score reflects this).

My main reservation is that the improvement in performance is only minor. That said, since it comes at little extra computational cost, it is certainly not an argument to not use it.

Related to the above comment, the deltas reported in Tables 4 and 5 are against the Reported accuracies from the source rather than the author’s discovered accuracies from their experiments. This seems to show their method(s) in a better light and so is a bit disingenuous.

Some minor comments:
-	Remove “the” (“the maliciously ...”) in first sentence of Introduction. Same at the start of the Ethics Statement
-	In Algorithm 1, n is undefined
-	“be integrating” should be “by integrating” in caption for Table 2
-	Are there missing bold highlights on some of the data in Table 3?
-	The Ethics Statement section is not an ethics statement. It is more of a summary of the paper.
-	In the Ethics Statement: a few grammatical corrections
    o	“effects of” should be “effects on”
    o	“and may promote new adversarial attacks” (i.e. add “and”
-	The second line of the two line equation in Annex B is corrupted (norm)
-	In Annex C, discussion of the MT attack refers to 9 classes. More general to say “all other classes” of course.


**Summary Of The Paper:**

This paper proposes both a Basic and a Model-Agnostic Meta-Attack, designed to automatically produce more effective update directions during adversarial example generation than standard or hand-crafted evasion attacks against deep neural networks.

**Summary Of The Review:**

The paper provides only small improvements in performance. But since this is at minimal computational cost, this is not necessarily an issue. Given the importance of effective adversarial example generation in properly assessing the risk in usage of ML solutions, any contribution to improving the state of the art is welcome.

The authors draw a distinction to a similar Learning to Learn approach from 2020, establishing that though their approach is not therefore strongly novel, is distinct and has merit in its own right.

Hence I would recommend this paper for publication.

---

> ### Author Response · Authors · 2021-11-17
> **Thank you for the supportive review**
>
> Thank you for the supportive review. We have uploaded a revision of our paper. Below we address the detailed comments. We hope you may find our response satisfactory and increase the score accordingly.
>
> ***Question 1: The performance improvement is minor***
>
> Developing methods that can evaluate adversarial robustness accurately and reliably is of profound importance. However, the progress of the current attack methods for robustness evaluation is slow. For example, the differences between MT, AA, and ODI in Table 4 are very small. Compared to the existing methods, our method not only achieves acceptable improvements for robustness evaluation against the 12 SOTA defense models, but also provides a plug-and-play module (learned optimizer) with good adaptiveness (see Table 2), which may benefit future adversarial attacks if new improvements have been made in different aspects.
>
> ***Question 2: The deltas reported in Tables 4 and 5***
>
> Thanks for the suggestion. In the revision,  we further show the improvements of our method over the best attack.
>
> ***Question 3: Minor comments***
>
> Thanks for the comments. In the revision, we fix the typos and rewrite the Ethics Statement.

---

### Official Review · Reviewer_Xe9v · 2021-10-24

**Correctness:** 4
**Technical Novelty And Significance:** 2
**Empirical Novelty And Significance:** 2
**Recommendation:** 6
**Confidence:** 4

**Main Review:**

I think the proposed adversarial attack algorithm makes a nice contribution to the adversarial machine learning community, and this method can be considered as the main evaluation for a newly proposed defense model.

The strengths are listed as follows:

1: The proposed adversarial attack has less computational cost compared with other state-of-the-art attacks (As shown in Table 4).

2: The proposed attack has outperformed other strong attacks under 12 robust defense methods.

3: The proposed attack can be applied to a large-scale dataset (ImageNet as shown in Table 5)

The weakness of the paper is the lack of novelty. Meta-learning-based adversarial attacks have been considered before and using RNN as an optimizer has also been considered. The contributions of this paper are those improvements made to improve the performance.



**Summary Of The Paper:**

This paper proposes a model-agnostic meta-attack and achieves promising adversarial attack performance compared with state-of-the-art adversarial attack algorithms. The proposed algorithm overcomes many issues, such as the vanishing gradient problem, training instability problem, generalization to other defense models. The detailed experiments support the proposed method.

**Summary Of The Review:**

In the perspective of proposing an adversarial attack algorithm, the author has conducted detailed experiments to support the method. However, the contributions of this paper are the improvements to the existing framework.

Hence, I support a marginal acceptance of this paper.

---

> ### Author Response · Authors · 2021-11-17
> **Thank you for the supportive review**
>
> Thank you for the supportive review. We have uploaded a revision of our paper. Below we clarify the novelty of our work. We hope you may find our response satisfactory and increase the score accordingly.
>
> ***Question: Novelty***
>
> Although there are some related works on meta-learning and RNN optimizer in adversarial attacks, our work has its own merits. The novelty can be summarized as follows.
>
> - **Technically**, we proposed two techniques to improve and stabilize the training of the RNN optimizer, including the straight-through estimator to solve the vanishing gradient problem and the prior-guided refinement strategy to better leverage the prior information. Both techniques have not been adopted in the most related work [1] and could be useful for the general field of learning-to-optimize (L2O). Moreover, we further developed a model-agnostic training algorithm to improve the generalization ability of the learned optimizer. In the field of L2O, most methods address the generalization problem by designing better optimizer architectures and training techniques. To the best of our knowledge, we are the first to adopt the model-agnostic training algorithm to train the RNN optimizer, which can also be insightful for future research of L2O, while not limited to adversarial attacks.
>
> - **Empirically**, we showed the advantages of the learned optimizer over hand-designed ones in adversarial attacks. Our method not only achieves acceptable improvements for robustness evaluation against the 12 state-of-the-art defense models, but also provides a plug-and-play module (learned optimizer) with good adaptiveness (see Table 2) and generalization ability (see Fig. 3, Fig. 4, Table 3, and Table 5). On the contrary, the related work [1] did not show the advantages of attacks using the learned optimizer but focused on adversarial training, as discussed in Appendix A.
>
> Reference:
>
> [1] Xiong and Hsieh. Improved adversarial training via learned optimizer. ECCV 2020.

---

### Official Review · Reviewer_8AXD · 2021-10-26

**Correctness:** 3
**Technical Novelty And Significance:** 2
**Empirical Novelty And Significance:** 2
**Recommendation:** 3
**Confidence:** 3

**Main Review:**

Pros:
+ the usage of meta learning techniques is interesting to create better attacks
+ preliminary step ahead for the research in creating better adaptive attacks

Cons:
- results suggest a too modest improvement w.r.t. the computational effort for training the attack
- only adversarial training is considered

Minor:
- confusion between MAMA and BMA

The creation of automatic adaptive attacks is an important step ahead for the adversarial machine learning community.
However, the proposed technique is hindered by the following key issues.

**Results suggest a too modest improvement w.r.t. the computational effort for training the attack**
The results proposed in the paper show that MAMA and BMA reduces the robustness accuracy of the strongest automatic attack (AA) by a very low delta (Table 5), but the table do not show all the effort that is needed for optimizing the optimizer.
Hence, I am doubtful about the overall technique, since it behaves similarly to other strategies that do not need to be trained on any models.
Also, the deltas shown in Table 5 are computed against the original evaluation, and not the output of the best attack against the considered defense. This is misleading, since the discussion should revolve around the efficacy of the proposed attacks against the other ones, and the table seems to suggest an improvement that is not real.

**Only adversarial training is considered**
While it is true that adversarial training (AT) is becoming widely used, it is not the only defense that has been proposed, or that it will be proposed in the future.
Hence, how this method can adapt to a technique whose loss landscape is very different from an adversarially-trained one? Intuitively, more defenses should be added to the set used in the algorithm, but more experiments should be conducted to understand their effect on the meta learner. Or rather, this should be stated as limitations (that are not discussed).
Finally, the fact that this strategy only addresses AT should be mentioned in the abstract, since it is now misleading the reader into believing that this method can adapt itself to every defense.

**Minor comment: confusion between MAMA and BMA**
The authors might need to better describe BMA and MAMA, maybe by creating a better visual plot.
As it is now, the paper starts by describing BMA, and later MAMA, but the paper title is centered on MAMA. So, the discovery of BMA is a bit misleading while reading.


**Summary Of The Paper:**

This paper proposes an automatic approach for attacking classifiers, by approximating a possible adaptive attack that can take place on a newly-published defense. The methodology (MAMA) applies meta machine learning techniques, by training it on different defenses to let it grasp what might be a good signal to follow when attacking an unknown classifier. The authors then test this strategy and a naive version of it (that is, the one that is only trained on the attacks, and not fine tuned on a test set) against other attacks proposed in literature.
Results suggest a modest improvement against defenses hardened with adversarial training.

**Summary Of The Review:**

I opt for a rejection, since I struggle to see a clear contribution w.r.t. the literature. Other attacks that do not need any optimization have comparable efficacy to the ones scored by BMA and MAMA, but without all the effort that is needed for training the meta learner. Also, the table is reporting misleading results in the delta column, since it is computed against the original evaluation, and not w.r.t. the strongest attack against a particular defense.
Also, these techniques only adapt themselves on adversarial trained models, limiting the scope of the meta learner efficacy.

---

> ### Author Response · Authors · 2021-11-17
> **Thank you for the valuable review (Part 2/2)**
>
> ***Question 3: Only adversarial training is considered***
>
> In this paper, we focused on adversarial training not only due to the superior performance but also because previous attacks (e.g., AutoAttack, MultiTargeted attack) mainly study adversarial training. Although all defenses we studied are based on adversarial training, they possess different network structures, initialization weights, and defense strategies. For example, we also included a defense based on FGSM adversarial training. Therefore, the loss landscapes could be significantly different among these models, and the learned optimizer still face the challenges of possessing good generalization ability to unseen defense models.
>
> We agree that adding other types of defenses can make our methods more compelling. Thus we further conduct experiments to evaluate the performance of our methods (i.e., BMA and MAMA) on other types of defenses, including a certified defense -- CROWN-IBP [1] and a randomized defense -- JEM-1 [2].
> We compare BMA and MAMA with AutoAttack, which is the most effective baseline. We re-train the attack optimizer for each model in BMA and directly use the learned optimizer (i.e., the same as Table 4) in MAMA. To counter the randomness of JEM-1, we average the gradients over 3 runs (known as Expectation over Transformation (EoT) [3]). The results are shown below.
>
> | Method | AutoAttack | BMA | MAMA |
> | :----- | :-----: | :----: |  :----: |
> | CROWN-IBP | 33.71 | **33.60** | 33.64 |
> | JEM-1 | 7.40 | **6.35** | 6.57 |
>
> Our methods consistently achieve better performance than AutoAttack. Note that MAMA uses the attack optimizer trained on four adversarially trained models, but it still demonstrates better performance for other types of defenses, showing the great generalizablity. In the revision, we provide these results in Appendix E.5.
>
> However, our methods still have the same limitation as previous attacks that they cannot directly be used to attack defenses with obfuscated gradients. We need to design some adaptive attack strategies (e.g., EoT for randomized defenses) to apply our methods. In the revision, we discuss the limitation of our methods in the Ethics Statement.
>
> Reference:
>
> [1] Zhang et al. Towards stable and efficient training of verifiably robust neural networks. ICLR 2020.
>
> [2] Grathwohl et al. Your classifier is secretly an energy based model and you should treat it like one. ICLR 2020.
>
> [3] Athalye et al. Synthesizing robust adversarial examples. ICML 2018.

---

> > ### Comment · Reviewer_8AXD · 2021-11-18
> > **Thank you for the replay 2/2**
> >
> > Thank you for having conducted other experiments on certified defenses.
> > The fact that you averaged the gradient is going against the plug-and-play philosophy of the paper, so I am a little concerned about the results that are shown in this table (because now you have trained a meta-learner and then applied the mitigation that improves the attack, and it is not model-agnostic anymore).
> > Also, BMA is performing better than MAMA (that should be even more complicated and perfomative), but both are improving the performances of 0.1% and roughly 1% against the tested models.
> > Again, I am still concerned if it is worthy to wait hours in optimizing a meta-learner and then obtaining basically the same performances than other SoA attacks.

---

> > > ### Author Response · Authors · 2021-11-20
> > > **Thank you for the further comment**
> > >
> > > Thank you for the further comments. Our proposed method MAMA can be model-agnostic only for defenses without obfuscated gradients, as we clarified in the Ethics Statement. Actually, none of the baseline attacks adopted in this paper (e.g., AutoAttack) can be plug-and-play for defenses with obfuscated gradients (e.g., the randomized defenses).
> > >
> > > For the randomized defense JEM-1, AutoAttack also adopted the average gradients over 3 runs to craft adversarial examples, and we used the same strategy in BMA and MAMA for a fair comparison.
> > >
> > > For the performance of BMA and MAMA, BMA trains a new optimizer while evaluating a new defense, and MAMA directly adopts the learned optimizer as same as that in Table 2 to evaluate the new types of defenses, thus BMA can obtain better performance.

---

> ### Author Response · Authors · 2021-11-17
> **Thank you for the valuable review (Part 1/2)**
>
> Thank you for the valuable review. We have uploaded a revision of our paper. Below we first dispel the confusion between BMA and MAMA, and then address other comments for better clarity. We hope you may find our response satisfactory and increase the score accordingly.
>
> ***Question 1: Confusion between BMA and MAMA***
>
> In BMA, we train the attack optimizer parameterized by a recurrent neural network (RNN) to maximize the objective function in Eq. (5). This optimization problem is usually solved by vanilla gradient ascent for each defense model. However, the learned optimizer may not possess a good generalization ability for attacking unseen defense models. Therefore, we further develop a model-agnostic training algorithm to train the attack optimizer, which is named MAMA. The learned optimizer by MAMA can directly be used to attack other defenses **without further re-training or fine-tuning**. Note that the major difference between BMA and MAMA lies in the training algorithms for the RNN optimizer. BMA can be viewed as a simplified version of MAMA, while is also an essential building component of MAMA. Fig. 1 shows the overview of MAMA, in which the top-left plot shows the architecture of the RNN optimizer in BMA.
>
> ***Question 2: A too modest improvement w.r.t. the computational effort for training the attack***
>
> **Computational effort.** We need to clarify a potential misunderstanding that MAMA does **not** need to train the optimizer for each defense model. Based on Algorithm 1, we train an attack optimizer using a few models by the model-agnostic training algorithm (which takes about 5h by utilizing 4 defense models on a single RTX 2080-Ti GPU). Once trained, the learned optimizer can directly be applied to evaluate other unseen defense models without further re-training or fine-tuning. Actually, we train a different optimizer by BMA for each defense, which obtains similar performance to MAMA as shown in Table 4. Therefore, MAMA is reliably used to evaluate the robustness of different defenses on various datasets, and is much faster than AutoAttack, as shown in Fig. 5.
>
> **A too modest improvement.** Developing methods that can evaluate adversarial robustness accurately and reliably is of profound importance. However, the progress of the current attack methods for robustness evaluation is slow. For example, the differences between MT, AA, and ODI in Table 4 are very small. Compared to the existing methods, our method not only achieves acceptable improvements for robustness evaluation against the 12 state-of-the-art defense models, but also provides a plug-and-play module (learned optimizer) with good adaptiveness (see Table 2), which may benefit future adversarial attacks if new improvements have been made in different aspects.
> In the revision, we further show the improvements of our method over the best attack to be not misleading.

---

> > ### Comment · Reviewer_8AXD · 2021-11-18
> > **Thank you for the reply 1/2**
> >
> > First of all, thank you for the reply, but I still have some issues with the comments.
> >
> > **Performance issues**. Thank you for better clarifying the difference between BMA and MAMA, but I am still doubtful on the fact that these attacks need to be trained (and hence, maintained), while other strategies are really plug-and-play (and this is not, since it needs training). I struggle to see the benefit w.r.t. other techniques of the SoA.
> >
> > **Modest improvement**. I still see a modest improvement, as reported also now clearly on the paper (thank you for the addition, I think it still can be improved because the $\Delta$ and $\Delta^\prime$ are still difficult to read, but I am ok with the edit), linked to the fact that the methodology is not plug and play (because it needs training), and the improvement is less than, on average, 0.1% (meaning that it helps finding 1 adversarial examples more than other really plug-and-play methods every 1000, but only after having trained the algorithms for hours). I agree to the fact that this field is evolving right now, but the cost of improving of few points on massive datasets is just a modest fine-tuning of proposed methods.

---

> > > ### Author Response · Authors · 2021-11-20
> > > **Thank you for the further comment**
> > >
> > > Thank you for the further comments. We would like to clarify some potential misunderstandings.
> > >
> > > 1) **MAMA is plug-and-play**. As we clarified in our initial response, MAMA only needs to train the attack optimizer once, and then it can be used for the evaluation of all defenses based on **generalization**. Just like SGD, MAMA can be regarded as a preferable optimizer in the robustness evaluation of defense models **without the necessity of updating optimizer parameters**.
> > >
> > > 2) **MAMA is more efficient than AutoAttack**. MAMA takes about 5 hours for training on a single RTX 2080-Ti GPU, and about 5 hours for evaluating a defense model. As a comparison,  AutoAttack needs more than 12 hours to evaluate a defense model. The training time of MAMA is even shorter than the inference time of AutoAttack. Furthermore, if we aim to evaluate the robustness of $N$ defense models, the average time required for each model is about $(5/N + 5)$ hours, whereas the average time of AutoAttack should be over 12 hours.
> > >
> > > Totally, compared with AutoAttack, MAMA not only achieves better performance, but also runs faster. Although the improvement is not very significant, the idea of using an automatic optimizer is first successfully applied to evaluate the robustness of defense models, and is orthogonal to previous approaches. It indicates that our method can be integrated with other techniques, and we showed the good adaptiveness of our method (see Table 2).

---

> > > > ### Comment · Reviewer_8AXD · 2021-11-25
> > > > **A clarfication on my comment**
> > > >
> > > > When I specify that I have an issue with the modest improvement of the proposed technique, I am not only referring to the AutoAttack procedure (which is known to be slow, since it is applying a lot of different attacks in a row), but also related to the others (like MT and ODI).
> > > > Hence, I totally agree that it is unbareable to wait that many hours for AutoAttack to conclude, but the other attacks are still a lot competitive with the approach. Maybe there are other dimensions where MAMA / BMA are better than these techniques as well?

---

> > > > > ### Author Response · Authors · 2021-11-26
> > > > > **Thank you for the further comment**
> > > > >
> > > > > Table R1 further shows the results of different attacks averaged over the 12 defenses in Table 4. It can be clearly seen that the performance gap between MT, ODI, and AutoAttack is less than 0.1\%. Compare with these attacks, we obtain a 0.1\% improvement, which is **still meaningful** due to the essential requirement for **worst-case** robustness evaluation.
> > > > >
> > > > > Although AutoAttack is much slower and averagely brings about less than 0.1\% improvement compared with MT and ODI, it is still the **most popular** benchmark right now due to its better performance. Therefore, MAMA can also serve as **a useful benchmark** for robustness evaluation due to the improved results.
> > > > >
> > > > > Overall, MAMA is more effective and much more efficient than AutoAttack, while the paradigm of learning an optimizer in MAMA is also **complementary** with existing attacks, which is a characteristic that other methods do not have.
> > > > >
> > > > >
> > > > >
> > > > > | PGD (2018) | APGD (2020) | MT (2019) | ODI (2020) | AA (2020)  | MAMA (2021) |
> > > > > | :-----: | :-----: | :----: |  :----: | :-----: | :-----: |
> > > > > | 56.27 | 55.01 | 53.50 | 53.43 | 53.41 | **53.31**|
> > > > >
> > > > > Table R1. Averaged accuracy of 12 defense models against different adversarial attacks.

---

### Decision · Program_Chairs · 2022-01-20

**Decision:**

Reject

**Comment:**

This paper introduces a "model agnostic" attack to evaluate the robustness
of machine learning models, specifically focusing on adversarial training
style defenses. The attack uses a RNN optimizer and meta-learning to achieve
this goal.

The reviews are mixed. The two more positive reviewers appreciated the
technical ideas of the attack, found the paper well written, and noted
the results were stronger than prior attacks.

However the more negative reviewer raises valid concerns around (a) the
magnitude of the contribution compared to prior attacks, and (b) to what
extent this attack will be useful more generally.

Starting with the first point (also raised by the other reviewers) the
total contribution of this paper is to improve attack success rates by
~0.1% compared to the best prior attacks. This is a fairly limited total
gain, especially because this technique requires much more sophisticated
attack techniques.

More fundamental is the question if this attack is useful to the community.
I tend to agree with reviewer 8AXD here that this contribution is rather
limited for two reasons:
1. As the authors acknowledge, the attack is most effective for adversarial
  training techniques, or others that don't make the gradient hard to
  optimize. This limits the attack to a smaller subset of defenses.
2. Complexity complicates attack evaluations. It's hard enough to get an
  attack working in the first place, and this paper has to use some fairly
  sophisticated tools to just get the attack marginally better than
  prior attacks that are (much) simpler. It's not clear that we would
  expect authors of future defenses to be able to get as good results,
  when prior methods are much simpler to apply.

And so on the whole, this paper's main contribution is improving attack
success rate by a small amount, with a fairly complex method, that only
applies to a class of defenses that don't make gradient descent difficult.
So while there is nothing outright wrong with this paper, in its current
form it seems to be adding unnecessary complexity to achieve something
that can already be done with existing techniques.